# The Effect of Dragon Boating on the Quality of Life for Breast Cancer Survivors: A Systematic Review

**DOI:** 10.3390/healthcare12131290

**Published:** 2024-06-27

**Authors:** Igor Herrero-Zapirain, Sergio Álvarez-Pardo, Arkaitz Castañeda-Babarro, Adrian Moreno-Villanueva, Juan Francisco Mielgo-Ayuso

**Affiliations:** 1Department of Health Sciences, Faculty of Health Sciences, University of Burgos, 09001 Burgos, Spain; igorjudo81@gmail.com (I.H.-Z.); jfmielgo@ubu.es (J.F.M.-A.); 2Faculty of Health Science, University Isabel I, 09003 Burgos, Spain; adrian.moreno@ui1.es; 3Health, Physical Activity and Sports Science Laboratory, Department of Physical Activity and Sports, Faculty of Psychology and Education, University of Deusto, 48007 Bilbao, Spain; arkaitz.castaneda@deusto.es

**Keywords:** cancer, rowing, quality of life, physical activity

## Abstract

Physical activity improves breast cancer-related symptoms in women and decreases cancer-related mortality. The main objective of this systematic review is to synthesize and analyze the evidence of the effect of dragon boating on the quality of life of female breast cancer survivors. A systematic review based on the PRISMA method was conducted using four databases (Web of Science, Scopus, Cochrane and Pubmed). The search phrase used was “Breast Cancer” AND “Dragon Boat” AND “Quality of Life”. The search was conducted in June 2024. The PEDro method was used to ensure the quality of the publications. A total of 77 articles published until 2024 were selected, of which 10 met the inclusion criteria of assessing the application of dragon boating and that used a validated instrument to assess quality of life. There is no homogeneity in terms of the instrument used to measure QOL. The SF-36 was the most commonly used, followed by the FACT-B and the EORTC QLQ-C30. Five out of ten articles compared the improvement in quality of life between dragon boating and other physical activities, while 6 out of 10 analyzed the pre–post effect of dragon boat use. Dragon boating is a physical activity alternative that improves the quality of life of breast cancer survivors and reduces the symptomatology caused by the disease and its treatments. As dragon boat programs are applied over a longer period of time, the improvements in quality of life are greater. When compared with other types of physical activity, dragon boating does not show significant differences that position it as a better option for this population.

## 1. Introduction

Cancer is one of the biggest public health problems in the world, being in some developed countries the main cause of death; in countries like the United States, 1670 people die every day because of cancer [1]. Breast cancer (BC) is the most common cancer affecting women worldwide, where the incidence is increasing by 0.5% each year since the mid-2000s, with an estimated 2.26 million registered cases in the last year [1], above colorectal cancer and lung cancer [2]. In the year 2023 in Spain, it is estimated that there will be approximately 35,000 new diagnoses of BC following the rising trend [3]. The main cause of this increase in incidence is the decline in healthy lifestyles. Despite the high incidence, the 5-year survival rate for BC diagnosed at stage I is over 99%, 93% for stage II, 75% for stage III and 29% for stage IV, this rate being lower in black women and in underdeveloped countries [4]. One of the reasons why there is a high survival rate when there are so many new cases of BC each year is the increase in research, which includes measurements of physical activity, the level of sedentary lifestyle, and physical capacity related to healthcare, providing a more complete assessment of the patient, leading to the implementation of physical activity programs that are completed during the development of the disease, reducing the risk of death from this pathology by 40% [5].

However, breast cancer survivors (BCSs) experience difficulties and complications both during and after the disease, including lymphoedema [6,7,8], upper limb stiffness or lack of mobility [9,10], inability to perform daily living tasks [11], insomnia [12] and fatigue [13,14], as well as psychological affectations such as depression, anxiety [15] and fear of recurrence [16], among others, which contribute to a decrease in the quality of life (QOL) of these women and consequently their survival [17]. The study of QOL in BCSs is a determining factor for health professionals to better identify people at a high risk of recurrence or mortality, and on the other hand, there are certain modifiable aspects of QOL that predict survival, and therefore, there is the possibility of interventions to reduce the risk of recurrence or death, such as physical activity [17].

Different guidelines have been created for physical activity (PA) to improve prognosis, QOL, physical well-being, cancer-related symptoms and survival [18]. Among the PA options, dragon boat (DB) racing is a sport modality that has been gaining popularity among practitioners worldwide in recent years due to its specificity for this population [19]. DB is a rowing sport in which a boat is propelled by 10–20 crew members distributed on both sides of the boat, with a coxswain and a drummer setting the paddling rhythm [19,20]. Participants paddle unilaterally with a single-bladed paddle from a seated position. DB is incorporated in the official calendar of the International Canoe Federation (CIF), with a special category called DB12 BCS, in which 100% of the participants must prove to be BCSs. Due to the championships and events that take place around this sport, its popularity has increased within this population, offering not only the opportunity to participate in the sport but also the chance to participate in competition events around the season.

The benefits of DB include an increasing time spent on PA, improving team cohesion, a growing sense of belonging to a group, and improving physical and emotional well-being [21], which is related to an increase in participants’ perceived QOL [19,21,22,23,24,25,26]. Some authors argue that the QOL of DB participants is higher than that of the general population [27,28,29]. Other authors find no differences in the QOL of DB practitioners over the duration of a study [9,21,22,30]. These differences could be due to the fact that the authors have used different parameters when establishing the duration of the study, aspects such as days/week and hours/week of PA and the measurement instrument, so that no homogeneity is found between the studies. This leads us to ask whether DB is a sport that improves QOL more than other PA options such as walking or resistance training [31] and whether there is a protocol of time of practice or hours of application of the sport per week that ensures the positive effect of the sport.

The aim of this systematic review is to answer the question, how does DB influence the QOL of BCSs? At the same time, we intend to determine which workout protocols, in terms of the total duration of the training session, both in length and in the ratio of hours/day or days/week of application, are the most effective in obtaining improvements in QOL. Finally, we want to compare whether the choice of one test or another to measure the QOL conditions the result of the improvement in QOL in DB paddlers. To our knowledge, this is the first systematic review of the QOL in BCSs who practice DB. The findings may provide relevant information to establish guidelines for PA in BCSs whose main option is DB.

## 2. Methodology

### 2.1. Literature Search Strategies

This article is a systematic review that studies the effects that DB produces on the QOL of BCSs. An evaluation of the methodological quality of the study was carried out by three authors.

The following databases were used to perform a structured search: Pubmed, Web of Science and Scopus. These high-quality databases ensured adequate bibliographic support, and the search was completed without limiting them to any specific year, including results up to and including 1 May 2023. The search phrase was as follows: “Dragon Boat” AND “Breast Cancer” AND “Quality of Life”. This review was performed in accordance with PRISMA guidelines.

### 2.2. Study Selection

The article selection was completed by screening the title and abstracts of the articles from the databases, and completed with a full-text review of all the articles that were considered potentially relevant, with a final analysis of their compliance with the inclusion criteria.

All titles and abstracts obtained were then cross-checked for any duplication or lack of actual studies on the topic.

### 2.3. Inclusion and Exclusion Criteria

For this systematic review, only scientific articles have been included. The articles were finally selected as a result of a database search using the following inclusion criteria: (I) the participants in the study were female breast cancer survivors, (II) the effects produced in the participants’ QOL were due to the practice of DB, (III) the articles were written in English and Spanish, and (IV) a measurement instrument was used to assess the improvement in QOL.

Exclusion criteria were also applied: (I) studies that did not involve the active participation of the subjects in DB activity, (II) studies that measured the application of another exercise program in DB paddlers, (III) studies merely descriptive of the sport of DB, (IV) studies that did not use any instrument or protocol to measure the effects of DB on QOL in breast cancer survivors, and (IV) studies not published in Pubmed, Web of Science or Scopus.

### 2.4. Quality Assessment and Risk of Bias 

Three separate authors assessed quality in terms of the methodology used, along with any risk of bias (I.H.-Z., S.A.-P. and A.C.-B.), and any lack of consensus was subject to assessment by a third party (J.M.-A. and A.M.-V.).

The methodological quality of the studies was assessed using the Physiotherapy Evidence Database (PEDro) scale. The scale scores the methodological quality of the studies and has a maximum score of 10. The scores were based on all the information available in the authors’ published version. A minimum of 3 out of 10 points is set as the minimum for inclusion in the review. The score for each study was determined by the authors (I.H.-Z., S.A.-P. and A.C.-B.). Disagreements were resolved by consensus or by a third and fourth reviewer (J.M.-A. and A.M.-V.).

### 2.5. Measuring Variables

The literature on the effects of DB on the QOL of BCS women was reviewed.

### 2.6. Data Extraction

Initially, inclusion/exclusion criteria applicable to all studies were used, and then data relating to the study source were extracted separately using a spreadsheet. These included the authors and year of publication, the topic of application, which variable they worked on, and the type of instrument used for data collection.

Any disputes were dealt with through discussion until an agreement was reached. For the purposes of review, we considered every study that had measured QOL using some measurement instrument that assessed the effects of active practice in DB by BCSs on QOL.

### 2.7. Synthesis

To analyze the articles included in the review, we took into account the sampler size, the population in which they were conducted, the instrument used for measurement, the type of intervention applied, the duration of the study, whether or not they had previous experience in DB and the effect of the intervention on QOL.

## 3. Results

The data extracted from the 10 articles included in the systematic review are explained in Table 1.

The characteristics of the articles included in the systematic review are detailed in Table 2.

### 3.1. Study Selecction

The literature search gathered a total of 77 articles (Figure 1), 19 of which came from Pubmed, 23 from Scopus, 3 from Cochrane and 32 from Web of Science. A total of 20 articles were eliminated as duplicates, leaving 57 articles, of which 3 were excluded by title and abstract selection. The remaining 54 full-text articles were assessed for eligibility. Therefore, 10 articles were included [9,21,22,27,28,29,30,31,32,33] for the following review.

### 3.2. Study Characteristics

Table 3 shows the general characteristics of the different articles selected in the current systematic review. The parameters of intervention time were analyzed, where 4 of 10 of the studies have an application of fewer than two months. The participation with or without lymphedema of the (N) is not specified in 6 out of 10 of the cases. On the other hand, the participation in races by the participants of these studies is 4 out of 10. In 6 of the 10 studies, the population shows previous experience in DB. The studies are divided into 3 of 10 in an observational study format and 6 of 10 in a pre–post format, these being the two most used formats. A control group was employed in 5 out of 10 of the studies, which the others did not have. Six different instruments were used in the selected studies, SF-36, FACT-B, and EORTC being the most commonly used instruments for measuring QOL.

Articles [9,27,28,29,30,32,33] obtained a good quality after analysis, with scores ranging from 6 to 8. The remaining articles [21,22,29] obtained a fair quality, with scores ranging from 4 to 5. It is worth noting that none of the articles included in the review scored points in item 3, which analyzes the concealed assignment (Table 3).

The extracted data were analyzed by grouping them into groups (length of the study, type of intervention, previous experience in DB, instrument used to measure QOL, effect of DB exercise on QOL), since they are considered the most important variables to take into account in these studies, given the importance of these variables in the future not only of the quality of life of the participants but also in the conditioning factor of the continuity of DB.

#### 3.2.1. Length of Study

In terms of duration, there are differences between authors. We found four articles with less than 8 weeks intervention, two of which only consist of one data collection. On the other hand, four studies use the duration of a season (6–7 months) as a time measure to observe changes in subjects’ QOL. Baseline measurements are taken during the first weeks of the start of the season, and endline measurements are taken either during the last weeks or after the end of the competitive period.

#### 3.2.2. Type of Intervention

With the exception of study [9], the rest of the studies have a prospective design, either observational, comparative or sequential. The way of obtaining data in most of the studies has been pre–post, in order to identify the changes produced by the application of DB between the first and second sample data collections. In the remaining two articles [9,28], data have been collected at a single point in time, by performing a comparative study with other subjects from other physical activities or with the general population.

#### 3.2.3. Previous Experience in DB

Most of the studies have used subject groups already trained in DB and with a predisposition to practice this sport. As the authors state, participants who had been practicing DB for more than one season showed a better baseline in their QOL [29,31]. In only one intervention, the groups have been randomly assigned [33]. The choice of whether or not to practice DB has come from the subjects of the study, because they had practiced it before, they have been interested in it themselves or they have been asked by the researchers to carry out the study. As can be seen from the articles, most of the participants had previously practiced dragon boating, which is an indication that it is a sport that generates adherence in its practice in this population in particular, providing long-term benefits.

#### 3.2.4. Instrument Used to Measure QOL

All the selected articles analyze the subjects’ QOL, but not all of them do so with the same instrument. As it can see in Table 2, the authors use different instruments to measure these data; among the most used, we find the SF-36 used in four studies, followed by the FACT-B and the EORTC QLQ C-30 used in three articles each. Five articles used more than one instrument to obtain the data.

### 3.3. Effect of DB Exercise on QOL

QOL tends to improve in subjects who practice DB. In 9 out of 10 of the studies, it was observed that practicing DB improves the participants’ QOL. In some cases, DB improves QOL more than physical activities like walking, pilates, yoga and resistance training [27,33]. In other studies, they state that DB improves QOL in a similar way to those who perform other physical activities, such as walking or resistance training [9,21,22]. As for the comparison between the DB groups and the control groups that do not perform PA, DB tends to significantly improve the QOL of the participants. On the other hand, DB participants show similar levels of QOL to the general population without BC.

The QOL tends to improve in those subjects who practice DB. In some studies, the QOL of dragon boaters improves more than that of those who perform physical activities like walking or resistance training [27,33]. In other studies, they claim that dragon boating improves QOL in the same way as other types of physical activity such as walking or resistance training [9,21,22]. QOL improves significantly when comparing dragon boaters to those who do not engage in any physical activity [27,32]. DB practitioners show similar levels of QOL to healthy women. DB proves to be a suitable option for BCSs with lymphedema, as well as for those without lymphedema [21,31]. Although BCSs with lymphedema do not have the same quality of life values as BCSs without lymphedema, dragon boating improves quality of life and is a physical activity option that demonstrates improvements in this population [27,28].

The appearance of lymphedema is lower in the groups that practice DB than in those who practice other types of physical activity or alternative therapies [27], or no activity at all [32]. When comparing the evolution of lymphedema before and after treatment, a significant improvement is observed in those women who practice DB, contributing to the improvement of QOL and reducing relative symptoms such as fatigue and insomnia [27,28], and late arm impairment [9], among others.

## 4. Discussion

The aim of this systematic review was to determine whether the practice of DB improves the QOL of BCSs who practice it. Furthermore, it aims to identify which aspects are relevant for this effect to occur. The main results indicate that DB improves the QOL of BCSs, improving the fitness level for daily functioning, causing them to feel better mentally and show greater self-acceptance and self-confidence [9,21,22,27,28,29,32,33]. The continued practice of DB is presented as a suitable activity because of its specificity for this population, improving the mobility and stiffness parameters of the operated-on arm [28], decreasing fatigue [13] and creating the feeling of belonging to a peer group [34]. However, in spite of the QOL improving after a DB intervention, no significant differences are found between DB and other PA such as walking or resistance training [9,21,22], with the exception of one study, in which they point out that DB improves QOL more than other physical activities performed twice a week, which would position this sport as one of the main options for this population [27].

BCSs suffer from complications during the disease and during its treatment, such as conditions that limit the mobility of the upper limb like lymphoedema, joint stiffness or pain, which affect their ability to perform daily life tasks and therefore diminish their QOL [28,35]. Resistance exercises of the upper limb have been shown to improve those pathologies derived from BC [6,7,8,9,10,11,36], so DB is presented as an optimal exercise alternative for this population, improving muscular strength in both the upper and lower extremities [8]. On the other hand, side effects such as depression, fatigue or insomnia produced by BC and its treatments decrease, offering them hope and the ability to regain control of their life, both when conventional PA is performed [12,13,15,16,37], and when DB is used as a PA option [28,29,38,39]. Finally, the social part and the feeling of belonging to a group promoted by PA tends to diminish the negative effects of BC by improving the QOL of BCSs, [34,40,41] since paddlers recognize that they feel among equals and do not tend to compare themselves with their peers, which would lead to a worse state of mind in terms of their healthcare [8,19,39,42,43,44].

DB is shown to be a good physical activity option for BCSs. Continuous physical activity that lasts longer than 6 months shows a more significant improvement in QOL [45,46] than research with a shorter duration, as in the case of BCSs who engage in physical activity lasting a year or more, who significantly increase their QOL [47] by improving their physical and social functioning and decreasing their anxiety and depression [48,49]. In this review, the same occurs if we compare studies covering a full season of DB [27,32] to those where its application is limited to 8–12 weeks [21,22,31]; we observe that the improvements in the QOL of DB participants improves more than the control groups, both sedentary and those who perform other PA. We can then say that a continuous DB practice of at least one season’s duration produces greater improvements in QOL than shorter time applications in BCSs. Among the characteristics that DB has to produce this adherence to the sport, we have the acceptance and respect that the paddlers feel from the members of the team, which increases the feeling of belonging [20,24], and on the other hand, sharing the activity with women who have suffered from the same disease makes them feel in a comfortable environment where they can share information, concerns and experiences [19,20,21,22,26]. In addition, it is an outdoor sport in the natural environment, which is an attraction for physical activity [21,50,51].

However, when it comes to comparing which physical activity increases subjects’ QOL the most, there are discrepancies between authors. One author stated DB participants increase their QOL more than those who practice other PA [27,33]; however, another one asserts that DB has benefits for BCSs, but finds no differences between those who practice DB and other PA [9]. One of the characteristics that differentiate the two studies is the duration of the activity, the first one being of longer duration, covering a DB season.

In most of the studies included in this review, the BC populations analyzed had previously practiced DB [9,28,30,31,32], and in two other articles, it does not specify whether they previously started in DB [27,29]; however, both do figure out the effect on QOL during a full season of DB. It is understood, therefore, that the population reviewed in most of the articles is a physically active population prior to the studies, namely, the people included in the studies were mostly DB practitioners prior to data collection; this must be taken into consideration, since the positive effects produced by PA, in this case DB, are not as significant as those in a sedentary population starting PA [52,53,54]. On the other hand, studies where participants were new to the sport discipline have had an 8-week application [21,22], in which no differences were found between DB and walking activity. It is essential to give enough importance to practicing PA not only during the course of the disease but also after overcoming it, where the risk of recurrence is lower if you maintain an active life and if you have an adequate body weight where there is more muscular mass and less fat mass, this being essential for increasing survival—which is where DB can play an important role.

## 5. Conclusions

Physical activity has been heavily studied as a method of improving breast cancer survival and breast cancer-related symptoms. In summary, although some studies indicate that there may be improvement, the results are contradictory, and it is not possible to state with certainty that DB is a suitable activity to improve the QOL of BCSs by improving disease-related and treatment-related symptoms like lymphedema, joint stiffness, fatigue, insomnia and pain, although improving daily activities and sleep quality and belonging to a peer group have positive effects on QOL. DB is not the only activity that improves the QOL of BCSs, nor is it clear that it is a better option than other PA. As this is the first systematic review that analyzes DB as a vehicle for enhancing the QOL of BCSs, more research is needed to better understand the effects of DB on BCSs.

This systematic review is not without limitations. First, the heterogeneity of the intervention protocols of the articles included for analysis, together with the small number of articles included, makes it difficult to draw strict and rigorous conclusions; second, the BCSs in the included articles already practiced rowing before being diagnosed with BC, so we cannot extrapolate these data to a sedentary population that wants to initiate DB; and third, the need to live near the sea or near an area with water, where you can sail and practice this sport, makes it more difficult to perform compared to other physical activity options.

The practical applications of this study are to unify research analyzing the effect of DB in the QOL of BCSs and to serve as a starting point for future research.

## Figures and Tables

**Figure 1 healthcare-12-01290-f001:**
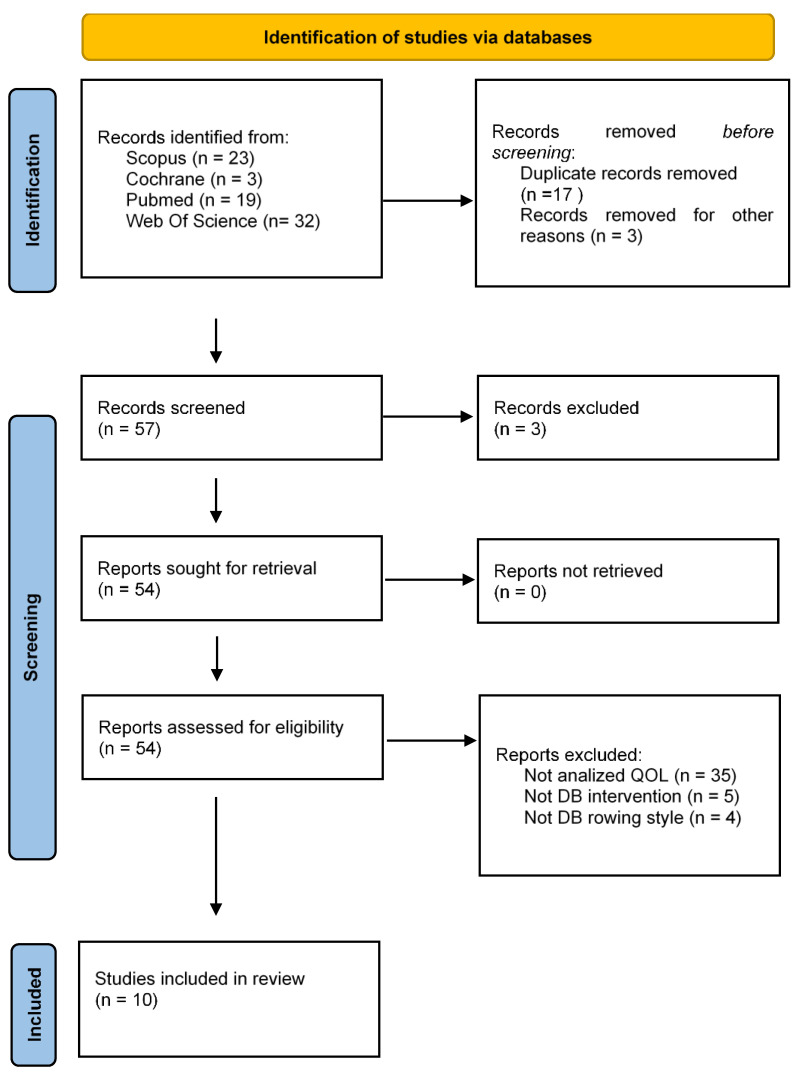
A flowchart of the process of selection, screening, suitability and inclusion of articles included in the systematic review. PRISMA 2020 guidelines.

**Table 1 healthcare-12-01290-t001:** Summary of studies included in systematic review investigating effect of DG on QOL in BCSs.

Autor and Year	Sampler Size	Population	Tool	Intervention	Follow Up (Duration)	DB Previous Experience	QOL
Melchiorri et al., 2017 [9]	N = 64 47 Unilateral mastectomy surgery G1= 15 Operated DB G2= 16 Operated FA G3= 17 BCS no PA + G4= 17 Healthy women	G1: BCS Operated 54.1 ± 5.8 TSS = 8.2 ± 2.3 G2: BCS Operated 53.7 ± 12.1 TSS = 8.6 ± 2.4 G3: BCS Non-Operated 60.4 ± 11.3 TSS = 7.2 ± 1.2 G4: Healthy women 51.1 ± 8.3	SF-36	Retrospective observational study	1 day	Yes	↑DB Operated > Non-Operated G1 > G2
Carter et al., 2010 [21]	N= 120 DB N = 68 WK N = 52	Cancer survivor 56% Breast cancer DB = 54.2 ± 9.7 TSD = 5.0 ± 5.7 WK = 58.2 ± 10.3 TSD = 6.6 ± 8.3	SF-36 FACT-B	Non-randomized intervention trial Pre–post	8 weeks 2 h weekly	No	↑DB ↑Walking DB = WK DB and walking = general population
Carter et al., 2012 [22]	N = 120 DB N = 68 WK N = 52	Cancer survivor 56% Breast cancer DB = 53.8 ± 9.7 TSD = 4.4 ± 5.1 WK = 58.2 ± 10.2 TSD = 7.3 ± 8.8	FACT-B SF-36	Non-randomized intervention trial Pre–post	8 weeks 2 h weekly	No	↑DB ↑WK DB = WK
Iacorossi et al., 2019 [27]	N = 100 participants 50 G1 = DB +6 month 50 G2 = PA biweekly	W BCSs 54% < 60 year	EORTC QLQ C-30	Observational study Pre–post	≥6 months	No specific	↑↑DB ↑PA DB > PA G1 > G2
Koehler et al., 2020 [28]	N = 748 Questionnaire	W BSCs who participate in DBR TSS = 9.0 ± 5.1	Lymph-ICF	Prospective, observational design	1 day	Yes	↑Lymphedema ↑Non-lymphedema Lymphedema non-lymphedema ↑DB ↑Overall average
Ray et al., 2013 [29]	116 1° Interview 100 2° Interview	W BSC 89% < 70 year TSD = 52% > 5 year	FACT-B FACIT-Sp12 FACIT-F + Semi-structured Interview n = 15	Mixed methods sequential explanatory design	Preseason– Postseason >6 months	Yes	↑↑DB
Denieffe et al., 2021 [30]	N = 49 analyzed All test = 26	W BCS and DB participants 54.5 ± 8.3	EORTC QLQ C-30 FACTI-F	Prospective observational pilot study	3 tests during 8 months	Yes	DB pre = DB post DB pre = DB mid DB mid = DB post
Culos-Reed et al., 2005 [31]	156 1° Interview 56 2° Interview	W BCS 52.89 ± 7.61	SF-12	Prospective observational Pre–post test	12 weeks 1 h week	Yes	DB = Overall average
Boeer et al., 2022 [32]	N = 98 DB = 28 CG = 70	Breast cancer survivor DB = 60.0 ± 9.2 TSS = 4.7 ± 4.0 CG = 62.5 ± 9.2 TSS = 6.7 ± 5.1	SF-36 EORTC QLQ C30	Prospective, comparative, longitudinally oriented case–control Pre–post	6 months Once weekly 1.5 h	Yes	EORTHC: ↑↑DB ↓Control SF-36: DB ↑ ↓↓Control DB > CG
Moro et al., 2024 [33]	N = 31 DB = 18 HP = 13	Breast cancer survivor DB = 56.40 ± 7.29 TSD = 7.33 ± 3.77 HP = 59.90 ± 9.67 TSD = 10.07 ± 7.38	SF-12	Parallel randomized trial	12 weeks 3 h weekly	No	↑DB DB > Control

BCS: breast cancer survivor; DBR: dragon boat race: DB: dragon boat; PA: physical activity: QOL: quality of life; ↑: improve QOL; ↓: decrease QOL; =: similar; ↑↑: improve significatively; ↓↓: decrease significatively; no similar; > more/less; CG: control group; WK: walking group; HP: home training program; TSD: time since diagnosis; TSS: time since surgery.

**Table 2 healthcare-12-01290-t002:** Characteristics of the articles included in the systematic review.

Characteristics	Variables	Nº of Articles
Length of the study	<2 months	4 studies (1, 3, 7, 8)
	From 2 to 6 months	2 studies (5, 10)
	>6 months	4 studies (2, 4, 6, 9)
Participants with lymphedema	Yes	3 studies (1, 2, 9)
	Not specified	7 studies (3, 4, 5, 6, 7, 8, 10)
Participants in races	Yes	4 studies (1, 4, 5, 6)
	No	2 studies (7, 8)
	Not specified	4 studies (2, 3, 9, 10)
Participants level	No previous experience in PA and DB	4 studies (6, 7, 8, 10)
	With previous experience in DB	6 studies (1, 2, 4, 5, 6, 9)
	Not specified	1 study (3)
Type of intervention	Pre–post	8 studies (2, 4, 5, 6, 7, 8, 9, 10)
	Observational study	2 studies (1, 3)
Instrument used to measure QOL	FACT-B	3 studies (4, 7, 8)
	SF-36	4 studies (3, 7, 8, 9)
	EORTC QLQ C-30	3 studies (2, 6, 9)
	FACIT-F	2 studies (4, 6)
	Lymph-ICF	1 study (1)
	SF-12	2 studies (5, 10)
	Semi-structured interviews	1 study (4)
Control group	Yes	6 studies (2, 3, 4, 8, 9, 10)
	No	4 studies (1, 5, 6, 7)

DB: dragon boat; PA: physical activity: QOL: quality of life.

**Table 3 healthcare-12-01290-t003:** Quality assessment of included studies (PEDro scale).

PEDro Items	Melchiorri et al., 2017 [9]	Carter et al., 2010 [21]	Carter et al., 2012 [22]	Iacorosi et al., 2019 [27]	Koehler et al., 2020 [28]	Ray et al., 2013 [29]	Denieffe et al., 2021 [30]	Culos-Reed et al., 2005 [31]	Boeer et al., 2022 [32]	Moro et al., 2024 [33]
1	1	1	1	1	1	1	1	1	1	1
2	0	0	0	0	0	0	0	0	0	1
3	0	0	0	0	0	0	0	0	0	0
4	1	1	1	1	1	1	1	1	1	1
5	0	0	0	0	0	0	0	0	0	0
6	1	0	0	0	0	1	0	0	0	0
7	1	0	0	0	1	1	0	0	0	1
8	1	0	0	1	1	1	1	0	1	1
9	1	1	1	1	1	1	1	0	1	1
10	1	1	1	1	1	1	1	1	1	1
11	1	1	1	1	1	1	1	1	1	1
Total	8	5	5	6	7	8	6	4	6	8
Quality	Good	Regular	Regular	Good	Good	Good	Good	Regular	Good	Good

## Data Availability

The data of the research are available in the tables of the manuscript or on request from the corresponding author.

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
