# Peer review of "The Effect of Dragon Boating on the Quality of Life for Breast Cancer Survivors: A Systematic Review"

_healthcare, 2024, doi:10.3390/healthcare12131290_

Round 1

Reviewer 1 Report

Comments and Suggestions for Authors

This systematic review was conducted to evaluate the impact of dragon boating on the quality of life of female breast cancer survivors.  The review analyzed data from Web of Science, Scopus, and PubMed. Of 135 articles published nine met the inclusion criteria. The studies varied in their QOL measurement tools, with the SF-36 being the most common, followed by FACT-B and EORTC QLQ-C30. The results suggest that dragon boating improves the quality of life and reduces symptoms in breast cancer survivors, with more significant benefits observed over extended periods. However, DB did not show significant advantages over other physical activities.

Abstract

The abstract is very well presented. Concise well written.

Introduction

It is well-presented and written.

Methods

The search is well explained. Would having precise strategies for each database in the appendix be helpful? Why is Cochrane not on the list of databases? How many authors performed screening? Was screening performed in duplicate as a quality assessment?

Inclusion exclusion criteria are well explained and clear.

A quality assessment was performed. I don’t think the results of this assessment belong in the methods section; I would move them to the results section.

Data extraction is well presented…. It would be good to specify who did it from the authors… how many?

I miss how the results were presented. Even if they're just descriptive, it's good to specify how the extracted data was analyzed…

Results

They are also well presented.

Instead of using percentages to present study characteristics, you can use numbers. For example, 5 out of 9 studies did this. Percentages don’t really help with this…

Again, in presenting the effect of Dragon Boad, use this number of studies as compared to the percentage…

Discussion is sound…

Author Response

Dear Reviewer, 

thank you very much for your comments on the manuscript, thanks to them we have been able to improve the manuscript and increase the quality of the content and writing of the paper.

Please find attached the answer sheet.

Kind regards

Reviewer 2 Report

Comments and Suggestions for Authors

Respected authors

You conducted a systematic review to test the effect of dragon boating (DB) on the quality of life (QOL) of female breast cancer survivors (BCS). This is an interesting topic although the results are not enough based on heterogeneity between studies. Overall, your manuscript needs some revisions.

- Please remove the subheadings from the abstract.

- Abstract, please add eligibility criteria in brief, and also, details regarding the synthesis of the results. 

- Why did you limit your search to 2001?

- Please add frequency or percentage for your results. For example what percent of the studies use the SF-36?

- Line 22, please use percentage and remove "some studies" here. 

- I think you must wright a conclusion with more caution. Your results are heterogeneous and you can not take a rigorous conclusion. 

- Please use similar words. effect? efficacy? or what? You used effect in the title and efficacy in the aim. It is not correct. 

- This statement is not correct; "The research was conducted using the PRISMA method". What does this mean? Please remove it. Also, please add only information regarding the search process under this heading. 

- Lines 106-7 and 102 are similar. Please remove one. 

- Lines 108-12 are related to study selection not search. Please follow PRISMA 2020 and refine all headings based on it.

- inclusion and exclusion criteria are different with study selection. Please refine it based on PRISMA 2020. 

- Please remove lines 136-141 from the method and replace them in the results section. Also, Table 1. 

- Please separate data extraction and data items from the synthesis methods. These are different steps of a systematic review. 

- Please change the 3.1. the subheading to the "study selection". 

- In my opinion, the reference style of the journal is different. Please check the journal guidelines and refine them. 

- Line 169, "Table 1 shows the general characteristics ...". Table 1 you used for quality appraisal. Please check your table. Table 3 must be table 1, then table 2, and table 1 renamed to table 3. Also, please mention Tables 1 and 2 at the end of section 3.2.

- Please remove column one from Table 3.

- Lines 340-47 are redundant. You mentioned limitation on the lines of 323-30. 

- Please write your conclusion with more caution. Also, you need more evidence in this regard to show the efficacy of this intervention. 

Author Response

(The authors gave the same response as above.)

Reviewer 3 Report

Comments and Suggestions for Authors

General Comments

Thank you for the opportunity to review your paper. In general, the topic of the paper is interesting and fits the scope of the journal. However, it is necessary to revise several aspects.

Specific Comments

Non-Published-Material: The PRISMA checklist is not reported correctly. Please replace the "X" with the “page and line number” where the items can be observed to simplify the review.

L12 – 26: Please make sure your abstract is written following the PRISMA 2020 for Abstracts checklist. Additionally, please remove abbreviations from the abstract.

L13: The objective of a systematic review should not be to "test". Please change it to "synthesize and analyse the evidence...".

L27: Three of your keywords are already listed in your title. If possible, modify them to increase the visibility of your review.

L89-96: The Systematic Review (SR) should articulate somewhere in the introduction or methods the main research question for SR as this question is traditionally used as a guide for other steps of the whole SR, such as establishing/developing keywords, eligibility criteria (inclusion and exclusion criteria), search strategy, etc. It’s a good practice to include research question for every SR.

L99-111: Provide more information in the methods section. For example, have PRISMA recommendations been followed? Have multiple authors participated in the data collection?

L102-106: Has a search for gray literature been conducted? This is to ensure a comprehensive searching and minimize risk of publication bias.

L105: The search has been ongoing for over a year. Please consider updating your search.

L112-113. The "Eligibility Criteria and Study Selection" section needs more detailed information. Were all types of resources, such as theses and books, included? and designs? Please indicate.

L136-140: Information on study quality analysis should be provided in the results section. Were the scores extracted directly from PEDro when available? Please indicate this.

L141:  Please improve the format of Table 1 and relocate it to the results section.

L155-163: Please reduce the text to avoid redundancy with the information provided in Figure 1.

Figure 1: Why have you adapted the PRISMA flow? If possible, please provide the original flow.

The format of Tables 2 and 3 also needs to be improved.

Provide more information about the study sample, including age, medication, reasons for dropouts, and time to diagnosis.

Provide clear information for intra-group and inter-group comparisons, either in tables or in the text.

L217: “other physical activities” – What were they? Please try to be as consistent as possible in the results section.

If possible, please clearly separate the practical applications of the study from the conclusions.

Please, check all abbreviations used

Author Response

(The authors gave the same response as above.)

Round 2

Reviewer 3 Report

Comments and Suggestions for Authors

Thank you to the authors for their effort in improving this review. I believe they have done an excellent job, but there are several points that remain unclear.

1. Non-Published-Material: The PRISMA checklist is not reported correctly. Please replace the "X" with the “page and line number” where the items can be observed to simplify the review.

Response: Thank you for pointing this out. We remake the PRISMA. See line 215.

I believe the authors have not correctly understood this comment. The original comment referred to the PRISMA checklist provided as "Non-Published-Material." This document is reported incorrectly.

5. The Systematic Review (SR) should articulate somewhere in the introduction or methods the main research question for SR as this question is traditionally used as a guide for other steps of the whole SR, such as establishing/developing keywords, eligibility criteria (inclusion and exclusion criteria), search strategy, etc. It’s a good practice to include research question for every SR.

Response: Thank you for pointing this out. We add research question in line 84.

In line 84, the authors report the objective of the review, but what is the question? This must be clearly formulated.

8. L105: The search has been ongoing for over a year. Please consider updating your search.

Response: Thank you for pointing this out. We update the research until 1 of June 2024. See line 100.

Line 100: The date of the last search does not match the date reported in the abstract.

L268: If you start the sentence by saying "The articles by ...", you must mention the authors. Were the scores extracted directly from PEDro when available? Please indicate this.

16. Provide more information about the study sample, including age, medication, reasons for dropouts, and time to diagnosis

Response: Thank you for pointing this out. We add more information about the study sample. See line 192.

I believe that the authors have made a significant effort to improve the table; however, they have not included information on dropouts, time to diagnosis, or medication.

Author Response

(The authors gave the same response as above.)
